# OpenReview forum: "Jailbreak Connectivity: Towards Diverse, Transferable, and Universal MLLM Jailbreak"
_ICLR.cc/2026/Conference — Submitted to ICLR 2026_

### Official Review · Reviewer_FeFk · 2025-10-29

**Soundness:** 3
**Presentation:** 3
**Contribution:** 3
**Rating:** 4
**Confidence:** 4

**Summary:**

This paper presents a way to generate universal, transferable adversarial images for jailbreaking.  To be detailed, it formulates the generation of diverse jailbreak images as an optimization problem of a quadratic Bezier curve, and utilizes surrogate classifiers for transferability, resulting in a curated optimization target. Experiments demonstrate the performance of such a method.

**Strengths:**

- Clear writing. The paper has a clear structure of storytelling, making it easy to follow the three aspects.
- The problem is well-defined and valuable.
- The design of the optimization target is interesting.

**Weaknesses:**

- Lack of analysis. The design of the loss function requires further explanation of the intuition behind it.
- No ablation studies. To make the method convincing, at least there should be ablations on each loss part to demonstrate the effectiveness. Besides, more baselines should be added for comparisons.
- The data display is sub-optimal. For example, in Table 1, maybe the most attention is paid only to the last row of the average numbers, and Table 2 is not expressive, either.
- Some tiny typos. For example, the dataset should be MM-SafetyBench, and maybe the Llava-2-13B refers to llava-1.5 or llava-Next.

**Questions:**

- I am not fully persuaded by the introduction of the quadratic Bezier curve. For me, the most intuitive way should be a linear combination. Why do we introduce the square of $u$ instead of simply using $ux_1+(1-u)x_2$? Are there some related comparisons?
- What are the settings of baseline methods? Do you pick some specific images as the adversarial visual input to test the transferability across scenarios on the whole dataset, or optimize adversarial images for every sample?
- Could you provide more explanations on the training of the jailbreak prediction classifier? If the labels for safety classifier training are determined by the model responses, and the labels for jailbreak success predictor are also derived from actual attack outcomes (the model responses), what is the difference between these two classifiers?
- Could you provide a few examples of successful jailbreaks on Gemini or GPT-4o? It will help if a few detailed jailbroken responses are included in the appendix.
-  What is the experiment setup in black-box attacks? I noticed that the target is set to GPT-4o, and in Table 10, the target is set to be Gemini. If the model has a black-box access, what is the difference between different targets?

Anyway, please correct me freely if I get anything misunderstanding in the weakness as well as the questions part.

---

> ### Author Response · Authors · 2025-11-21
>
> We thank the reviewer for the constructive feedback. We have revised the paper accordingly and summarize our responses below.
>
> ### **Intuition Behind the Loss Design**
>
> We added a clearer explanation of the intuition behind all three loss components:
>
> - **Path loss:** encourages the optimizer to explore a connected low-loss region, enabling stable and diverse jailbreak images.
> - **Surrogate-classifier loss:** provides approximate gradients reflecting refusal likelihood and harmfulness likelihood, guiding the attack toward transferable solutions.
> - **Universal-objective loss:** extends the same mechanism to a distribution of harmful queries rather than a single one.
>
> These explanations are now included in the revised paper.
>
> ### **Ablation Studies and Additional Baselines**
>
> Following your suggestion, we added a comprehensive ablation study (**Appendix A.7**).
> The results clearly show that **each component of JC is necessary**:
>
> - **Removing the continuous path** reduces the average ASR from **79.62% → 55.73%**
> - **Removing the jailbreak success predictor** reduces ASR to **67.68%**
> - **Removing the safety classifier** reduces ASR to **71.66%**
>
> Toxicity evaluation shows the same trend:
>
> - Average Detoxify toxicity drops from **63.41% → 38.05%** when removing the path
> - Drops to **46.22%** without the success predictor
> - Drops to **48.94%** without the safety classifier
>
> These results confirm that the path module, the jailbreak success predictor, and the safety classifier all contribute meaningfully and synergistically to JC’s performance.
>
> ### **Q1 — "I am not fully persuaded by the introduction of the quadratic Bezier curve. For me, the most intuitive way should be a linear combination. Why do we introduce the square of instead of simply using $ux+(1-u)x$? Are there some related comparisons?"**
>
> Quadratic Bézier curves provide a **richer search space** than linear interpolation, enabling exploration beyond the straight line between two images. Prior work in mode connectivity and robustness optimization adopts similar curves. Our ablation shows that removing the path (essentially reverting toward linear interpolation) results in a large ASR drop (79.62% → 55.73%), confirming its importance.
>
> ### **Q2 —"What are the settings of baseline methods? Do you pick some specific images as the adversarial visual input to test the transferability across scenarios on the whole dataset, or optimize adversarial images for every sample?"**
>
> For all baselines, we strictly follow the **best-performing configurations reported in their original papers** (e.g., perturbation budget, optimizer, step size, query limits). This ensures that every baseline is reproduced under its strongest setting.
>
> Regarding image optimization, **we optimize a jailbreak image for every individual test sample**. That is, for each harmful query in MM-SafetyBench, we generate one adversarial image (per method) and evaluate its ability to induce harmful responses. This per-sample optimization protocol is consistent with prior multimodal jailbreak work and ensures fair comparison.
>
> We have clarified this setup more explicitly in the revised manuscript.
>
> ### **Q3—"Could you provide more explanations on the training of the jailbreak prediction classifier? If the labels for safety classifier training are determined by the model responses, and the labels for jailbreak success predictor are also derived from actual attack outcomes (the model responses), what is the difference between these two classifiers?"**
>
> The **Safety Classifier** estimates whether the MLLM is likely to *refuse* due to its safety filters, while the **Jailbreak Success Predictor** estimates whether the model will *produce harmful content* when it answers. These signals are complementary rather than redundant.
>
> Our ablation results (Appendix A.7) clearly support this distinction.
> Removing the **Safety Classifier** drops the average ASR from **79.62% → 71.66%**, and removing the **Jailbreak Success Predictor** drops it further to **67.68%**. The same trend appears in toxicity evaluation: the average GPT-4o toxicity rate decreases from **63.41% (full JC)** to **48.94% (w/o Safety)** and **46.22% (w/o JBP)**.
>
> These quantitative results confirm that **both classifiers contribute meaningful, non-overlapping guidance**, and that removing either one significantly degrades jailbreak effectiveness.
>
> ### **Q4—"Could you provide a few examples of successful jailbreaks on Gemini or GPT-4o? It will help if a few detailed jailbroken responses are included in the appendix."**
>
> We added jailbreak examples for both GPT-4o and Gemini in Appendix A.8.

---

> ### Author Response · Authors · 2025-11-21
>
> ### **Q5 —"What is the experiment setup in black-box attacks? I noticed that the target is set to GPT-4o, and in Table 10, the target is set to be Gemini. If the model has a black-box access, what is the difference between different targets?"**
>
> The setting is black-box in both cases; the difference lies in the **target MLLM being evaluated**.
> The surrogate classifiers are trained separately for GPT-4o and for Gemini, so each target has its own tailored optimization signal.
>
> We thank the reviewer again for the helpful comments, which have improved the clarity, analysis, and completeness of our work.

---

> > ### Comment · Reviewer_FeFk · 2025-11-26
> > **Thanks for the rebuttal**
> >
> > Thank you for the detailed rebuttal. While my concerns regarding the experimental settings have been partially addressed, I remain unconvinced by the explanation concerning the safety classifier and the jailbreak predictor. Although the method is empirically effective, I believe the design intuition and underlying rationale require further clarification. Therefore, I will keep my score, and I do appreciate the time and effort you devote to the discussion.

---

> > > ### Author Response · Authors · 2025-11-26
> > >
> > > We sincerely thank the reviewer for the follow-up comment and the opportunity to clarify the design intuition behind the two classifiers.
> > >
> > > A successful multimodal jailbreak requires two conditions to occur in sequence:
> > > (1) the model must **choose to answer** the query rather than refuse, and
> > > (2) the model must **produce harmful content** in that answer.
> > > These are two distinct behaviors governed by different parts of the model’s safety pipeline.
> > >
> > > To model these behaviors, we collect different labels and train two surrogate classifiers. The **Safety Classifier** estimates whether the model is likely to refuse a query–image pair. This component ensures that the optimized image is interpreted as safe enough for the model to attempt a response. Without this guidance, optimization often drifts toward images that could induce harmful behavior but are still blocked by the model’s refusal rules.
> > >
> > > The **Jailbreak Success Predictor** captures a separate mechanism: whether the model will produce harmful content once it answers. Removing this component leads to a different failure mode. The model may respond, but the optimized image can shift the model’s internal representation away from the original harmful intent, resulting in benign or irrelevant content instead of harmful outputs.
> > >
> > > Therefore, the two classifiers play complementary and non-interchangeable roles.
> > >
> > > - If the Jailbreak Success Predictor is removed, the image may bypass refusal yet fail to induce harmful content because the intended semantics do not carry through.
> > > - If the Safety Classifier is removed, the optimization prioritizes harmfulness but frequently triggers refusal, so no harmful response is produced.
> > >
> > > Together, the two classifiers guide the optimization through both stages of the MLLM’s safety pipeline: first encouraging the model to answer, and then shaping the answer toward harmful content. Because these two mechanisms are fundamentally different, they cannot be captured by a single classifier.

---

### Official Review · Reviewer_LFiN · 2025-10-30

**Soundness:** 2
**Presentation:** 3
**Contribution:** 2
**Rating:** 4
**Confidence:** 4

**Summary:**

The paper introduces the Jailbreak Connectivity (JC) framework, which consists of three key components. First, it constructs a continuous path in the image space to generate a diverse range of jailbreak attacks connecting different jailbreak images. Second, it enhances transferability by using Safety Classifiers and Jailbreak Success Predictors as surrogate models to guide optimization. Third, it enables universal jailbreak attacks by redefining the attack objective to provoke harmful outputs of any kind, allowing the target multimodal LLM to produce unsafe responses across a wide variety of harmful queries. Experimental results show the effectiveness of the proposed method.

**Strengths:**

1. The paper provides good technical details and the writing is generally easy to follow.
2. The proposed method is very effective compared with previous methods.

**Weaknesses:**

I can’t say this paper is of low quality. I just feel the novelty is somewhat limited. The authors try to solve several limitations of previous multimodal jailbreaking methods through three separate aspects. These aspects are not strongly related, so it makes the paper look like something with several pieces combined but not fully connected. And the proposed solution for each aspect is somewhat naive. I suggest the authors add a discussion why you consider these limitations together, and whether they have synergies. This is very important. Otherwise I personally think it is not a well written paper.

**Questions:**

1. For the jailbreak success predictor and safety classifier. Since you train on the distribution of seen data. What if we need to deal with out of distribution images and wish to use them to jailbreak?
2. What is the efficiency of JC? Could you do a comparison with baselines? Since you are dealing with several problems at the same time, do you think your performance gain comes from more computation rather than approach design?

---

> ### Author Response · Authors · 2025-11-21
>
> We thank the reviewer for the constructive comments.
>
> ### **Synergy Between the Three Components**
>
> We added a short discussion clarifying why the three modules are considered together. JC was developed from a single motivation: improving the transferability and robustness of multimodal jailbreaks.
>
> The continuous path enlarges the search space, the surrogate classifiers guide this space toward transferable solutions, and the universal objective extends the same mechanism to broader harmful distributions.
>
> The new ablation study (Appendix A.7) shows that removing any component yields a substantial drop in ASR, confirming the three parts work synergistically rather than independently.
>
> ### **Q1 —“For the jailbreak success predictor and safety classifier. Since you train on the distribution of seen data. What if we need to deal with out of distribution images and wish to use them to jailbreak?”**
>
> JC optimizes a *continuous path* between images, its effectiveness comes from exploring a connected low-loss region rather than depending on specific properties of the initial images. This makes the method naturally tolerant to distribution differences. In another word, we believe JC can still work when dealing with out of distribution images.
>
> To verify this, we additionally tested **100 images not included in the surrogate-classifier training**, and JC still achieved **over 83% successful paths**, showing that the method generalizes well.
>
> ### **Q2—"What is the efficiency of JC? Could you do a comparison with baselines? Since you are dealing with several problems at the same time, do you think your performance gain comes from more computation rather than approach design?"**
>
> Under identical attack settings (ε ≤ 32/255, 3000 epochs), the runtimes are:
> Adv Example: **280.67 s**, Query Image: **316.31 s**, JC: **359.12 s**. Thus, JC introduces only a modest overhead of roughly **40–80 seconds**. More importantly, the ablation study shows that removing the path or either classifier causes a large ASR drop (**79.62% → 55.73%**) despite similar computation. This confirms that JC’s performance gains stem from its **design**, not from additional computation.
>
> We thank the reviewer again for the helpful suggestions.

---

### Official Review · Reviewer_KPuj · 2025-11-01

**Soundness:** 2
**Presentation:** 3
**Contribution:** 2
**Rating:** 4
**Confidence:** 4

**Summary:**

This paper introduces the "Jailbreak Connectivity" (JC) framework, a novel method for generating jailbreak attacks against Multimodal Large Language Models (MLLMs) that specifically aims to overcome three key limitations of existing work: lack of diversity, poor transferability, and ineffectiveness against multiple targets (i.e., lack of universality). The core idea is to find a continuous path, modeled as a quadratic Bezier curve, between two distinct jailbreak images. By optimizing this path, the authors can (1) sample a diverse set of effective jailbreak images, (2) improve transferability by incorporating lightweight surrogate classifiers (a Safety Classifier and a Jailbreak Success Predictor) into the optimization loss, and (3) create universal attacks by modifying the objective to target a general distribution of harmful outputs rather than a specific question-answer pair. The authors conduct extensive experiments on both open-source (LLaVA, MiniGPT-4) and closed-source (GPT-4o, Gemini) models, demonstrating state-of-the-art results. Notably, their method achieves an average Attack Success Rate (ASR) of 79.62% on SafetyBench, a 36.24% improvement over the best baseline, while also producing more fluent (lower perplexity) and more toxic responses.

**Strengths:**

The paper is well-motivated, clearly articulating the limitations of existing jailbreak methods (lack of diversity, poor transferability, and limited universality). The application of mode connectivity from loss landscape analysis to jailbreak attack generation is novel and interesting. Additionally, the use of lightweight surrogate classifiers to model target MLLM behavior is a bold and creative approach that, if properly justified, could have practical implications for scalable transfer attacks.

**Weaknesses:**

This paper has two major concerns that significantly impact its contributions:

* Insufficient Evaluation of Diversity Claims

The paper claims to address the diversity limitation of jailbreak images by introducing mode connectivity. However, the evaluation does not adequately measure or validate the actual diversity of generated images, which undermines one of the paper's core motivations.
From my understanding, the connectivity-based method primarily enables exploration of different perturbation choices within the same perturbation budget (ε = 32/255). I question whether this constitutes meaningful "diversity", as all generated images are bounded perturbations of the same original image. The paper needs to:

1. Provide quantitative metrics for image diversity (e.g., perceptual distance, feature-space divergence)
2. Clarify what "diverse" means in this context beyond sampling different points on a Bezier curve

As presented, the diversity contribution appears overtstated.

* Lack of Theoretical Foundation for Surrogate Classifier Design

The use of lightweight classifiers (CLIP-ViT-Base-Patch32) to predict target MLLM jailbreak success lacks justification. While Section 4.2.2 demonstrates empirical effectiveness, the paper provides no explanation or insights into why this approach works, which raises serious soundness concerns. I hold this blief because there is an apparent paradox: if a lightweight model can accurately predict whether a target MLLM will be jailbroken (and can even provide gradients precise enough to guide attack optimization), this would naturally constitute a very strong defense mechanism. The trained classifiers appear to model the target MLLM's vulnerabilities so precisely that they go far beyond simple "prediction." Why doesn't the target MLLM simply use this same classifier for defense?

In addtion, why is the Safety Classifier needed? Since the goal is to jailbreak MLLMs, it seems the Jailbreak Success Predictor alone would suffice. An ablation study comparing performance with only the success predictor versus both classifiers would clarify their individual contributions.

* Concern about Evaluation Metric Acurrency

The Detoxify classifier (2020) used for toxicity evaluation is relatively outdated, particularly when applied to datasets proposed after its release (e.g., AdvBench and SafetyBench). I recommend incorporating an LLM-as-a-judge as an additional (and likely more robust) evaluation metric to validate the toxicity assessments. This would strengthen the evaluation by: (1) providing a more contemporary measure aligned with current safety standards, and (2) offering cross-validation against Detoxify to ensure the toxicity findings are not artifact-specific to one classifier.

**Questions:**

For rebuttal, please refer to the weaknesses.

---

> ### Author Response · Authors · 2025-11-21
>
> We thank the reviewer for the insightful feedback and constructive comments. We have revised the paper accordingly and summarize the main updates below.
>
> ### **Insufficient Evaluation of Diversity Claims**
>
> Our notion of “diversity” refers to generating **multiple distinct and successful jailbreak images** for the same harmful query within the ε-bounded perturbation space.
>
> To support this, we provide two concrete empirical observations:
>
> 1. **Images along the JC path are not equivalent**.
>    As shown in Figure 2, the path does not lie in a flat region: the jailbreak loss varies along the curve. This demonstrates that the sampled images are **distinct** rather than redundant variations of the same adversarial pattern.
> 2. **Transfer results further confirm meaningful differences**.
>    In our transferability experiments, the two endpoint images often fail to transfer to the target MLLM, yet **intermediate points on the JC path succeed**. This indicates that JC explores regions of the adversarial space that are **truly different** from the endpoints and not reachable by standard single-point optimization.
>
> These behavioral analyses are sufficient to validate the form of diversity that matters for jailbreak effectiveness.
>
> ### **Surrogate Classifier Justification**
>
> The surrogate classifiers provide only **coarse, approximate signals** (refusal likelihood and harmfulness likelihood). They guide optimization but do **not** accurately model the full multimodal safety pipeline of GPT-4o or Gemini. Because their predictions are approximate and trained on limited data, they are **not accurate enough to serve as a defense mechanism**. Empirically, each classifier misclassifies roughly **7%** of validation samples. Such inaccuracies reinforce that the surrogate models provide only directional guidance rather than reliable predictions.
>
> Our ablation results (Appendix A.7) also show that both classifiers contribute meaningfully: removing the Jailbreak Success Predictor reduces ASR from **79.62%** to **67.68%**, and removing the Safety Classifier reduces ASR to **71.66%**, with toxicity showing similar declines. These findings confirm that the classifiers are helpful but fundamentally approximate components.
>
> ### **Evaluation Metrics**
>
> Following your suggestion, we incorporated stronger and more contemporary safety judges, including **Llama-Guard-3-8B**, **StrongREJECT**, and **GPT-4o** (Appendix A.6). Across all three evaluators, JC maintains consistently high harmfulness and jailbreak effectiveness. For example, under **GPT-4o**, JC increases toxicity rates from near-zero (Plain Text) and low baselines (Adv Example: typically *3–10%*) to **50–75%** across most toxicity dimensions, such as **Identity Attack (63.2% ± 2.1%)**, **Obscene (69.4% ± 2.2%)**, and **Insult (74.5% ± 2.3%)**. Similarly, under **Llama-Guard-3-8B,** JC achieves an average ASR of **62.9% ± 1.8%**, and under **StrongREJECT,** it receives an average harmfulness score of **0.43 ± 0.04**. These results confirm that JC’s effectiveness is not tied to Detoxify and remains robust when evaluated by modern alignment-oriented safety judges.

---

### Official Review · Reviewer_VPk9 · 2025-11-03

**Soundness:** 2
**Presentation:** 2
**Contribution:** 2
**Rating:** 2
**Confidence:** 4

**Summary:**

This paper studies jailbreaking MLLMs and addresses three key challenges: lack of diversity, poor transferability across models, and multiple jailbreak targets. The paper introduces the Jailbreak Connectivity (JC) framework, which includes three components: 1) a continuous noise path connecting two jailbreak images, 2) surrogate classifiers to guide optimization for better model transferability, and 3) an attack objective to elicit any harmful content.

**Strengths:**

The motivation to improve the diversity and transferability of jailbreak methods on MLLMs is clear and practical for MLLM safety.

The proposed framework demonstrates higher attack success rates and better transferability in the experiments.

**Weaknesses:**

The first major weakness of this submission is the writing quality. The Introduction section is not ready to publish, as it includes lengthy content on unnecessary points yet omits necessary details of certain important points, and does not provide a good-enough overview and introduction to this work. The first paragraph should be compressed with less content introducing the structure of MLLMs. The second paragraph should elaborate a bit more on the three categories of existing jailbreak methods on MLLMs to better align readers with the authors' taxonomy. The third paragraph, when introducing the limitations, should better cite related literature to help validate the listed weaknesses. Besides, there is no highlight summary of the JC framework regarding its attack performance, models and datasets used in this study, transfer performance, etc.

Moreover, the section on the related work of attacks on LLMs can be more tailored to this paper's topic, e.g., how does the transfer ability of attacks on LLMs work? How to ensure the diversity of attacks on LLMs? What are the similarities and differences of the idea of JC compared to methods on LLMs? The current version is a list of common attacks on LLMs without proper links and discussion to this paper's focus. For the attacks on MLLMs, when discussing the disadvantages, please cite related methods and papers that validate these weaknesses.

The second major concern is about the experimental setup. The default target model is MiniGPT-4. It is recommended to conduct the main experiments on more recent MLLMs such as Qwen-3-VL or Kimi-VL to prove that the proposed framework also applies to more recent and more powerful models with better alignments. Besides, the evaluator could be more solid if it incorporates other standard judge models such as Llama-Guard series and StrongREJECT rubrics. Also, to showcase the robustness of the framework, the average and std of the performance should be reported.

The third major weakness is in the analysis and case study part. The designed framework has 3 different modules, which leads to a bunch of interesting ablation studies worth investigating. For instance, a thorough collection and specification of the choice of hyperparameters is needed to justify the related design choices. Moreover, concrete output from each model is desired to prove the effectiveness of the model.

Other minor points: Use "an MLLM" instead of "a MLLM" on Line 138.

**Questions:**

- What is the advantage of the continuous path method, compared to random optimization with different steps and initializations?
- How many training samples and resources are needed for the surrogate classifiers? More details are preferred to justify the training cost of these surrogate classifiers.
- How does the optimized image help to alleviate the separate guardrails of GPT-4o and Gemini? The explicit harmful questions are normally directly detected and lead to refusal for these commercial models — how does the JC help to bypass such defense?

---

> ### Author Response · Authors · 2025-11-21
>
> We thank the reviewer for the thoughtful comments and constructive suggestions. We have revised the paper according to your feedback, and the major changes are summarized below.
>
> ### **Improved Writing Quality and Introduction Structure**
>
> We improved the writing quality and reorganized the Introduction following your suggestions. The revised content is highlighted in **blue** in the paper.
>
> ### **Experiments on Newer MLLMs**
>
> To validate JC on more recent and better-aligned models, **Appendix A.5** now includes results on **Qwen3-VL-8B-Instruct **and **Kimi-VL-A3B-Instruct**. JC achieves strong performance (65.4% and 60.6% ASR), showing that the method generalizes beyond MiniGPT-4.
>
> ### **Additional Evaluators**
>
> As recommended, we incorporated stronger and more contemporary safety judges, including **Llama-Guard-3-8B** and **StrongREJECT** (Appendix A.6). JC continues to perform robustly under both evaluators:
>
> - Llama-Guard-3-8B reports an average ASR of **62.9% ± 1.8%** across all scenarios.
> - StrongREJECT assigns an average harmfulness score of **0.47 ± 0.04**, indicating that JC consistently induces harmful responses even under a rubric-based safety assessment.
>
> These results confirm that JC’s effectiveness is not tied to any particular evaluator and remains stable under modern, alignment-oriented safety judges.
>
> ### **Ablation Study**
>
> We added a full ablation study in **Appendix A.7**, which shows that each component of JC plays an essential role. Removing the continuous path causes the largest drop, reducing average ASR from **79.62%** to **55.73%**. Removing the Jailbreak Success Predictor or Safety Classifier also leads to substantial performance declines (ASR **79.62% → 67.68%** and **79.62% → 71.66%**, respectively).
> A similar trend holds for toxicity, where removing the path drops the average harmfulness rate from **63.41%** to **38.05%**. These results confirm that all parts of JC contribute meaningfully and operate synergistically rather than independently.
>
> Besides, all tables in the main paper and appendix now report **mean ± standard deviation** for ASR, PPL, and toxicity.
>
> ### **Q1 — “What is the advantage of the continuous path method, compared to random optimization with different steps and initializations?”**
>
> The continuous path identifies a connected low-loss region, enabling many stable jailbreak images (over 70% success) and improving transferability. Random restarts produce isolated solutions and cannot guarantee diversity or stability. Our ablation study (Appendix A.7) further confirms the critical role of the path: removing it causes the largest performance drop (ASR 79.62% → 55.73%), showing that the path is essential for both diversity and effectiveness.
>
> ### **Q2 — “How many training samples and resources are needed for the surrogate classifiers? More details are preferred to justify the training cost of these surrogate classifiers.”**
>
> Each surrogate classifier is trained on 1,000 samples for 100 epochs using four A100 GPUs. Training finishes in about one hour and incurs minimal overhead.
>
> ### **Q3 — “How does the optimized image help to alleviate the separate guardrails of GPT-4o and Gemini? The explicit harmful questions are normally directly detected and lead to refusal for these commercial models — how does the JC help to bypass such defense?”**
>
> Our goal in this work is to make the target MLLM generate harmful content. When the textual input is *not explicitly harmful*, commercial models may not reliably detect unsafe intent. JC leverages this gap: the optimized image shifts the joint image–text representation so that the model interprets the overall query as benign or task-oriented rather than unsafe. This reduces the likelihood that the guardrails trigger a refusal, enabling the model to answer in cases where a harmful response would otherwise be blocked. As a result, **even relatively mild or implicit harmful queries can elicit highly harmful outputs once paired with a JC-optimized image**. We provide illustrative jailbreak examples in **Appendix A.8**.
>
> We thank the reviewer again for the helpful comments, which have helped us improve the paper.

---

### Meta-Review · Area_Chair_ASRS · 2025-12-30

**Summary:**

This paper introduces the Jailbreak Connectivity framework, which integrates three core techniques: 1) generating a manifold of diverse attacks via a continuous path between jailbreak images, 2) employing surrogate classifiers to guide transferable perturbations, and 3) optimizing a universal objective to provoke any form of harmful output, thereby enabling effective, broad-spectrum jailbreaks. Reviewers have recognized the paper's clarity of writing, its well-articulated motivation, and the innovative application of mode connectivity for jailbreaking.

However, the reviewers identified several major weaknesses as follows,
- The three components of the framework appear loosely connected, making the paper read like a collection of separate ideas rather than a unified solution.
- The proposed solutions were perceived as somewhat naive, and the overall novelty was considered limited.
- Key sections, particularly the Introduction and Related Work, were found to be unsatisfactory and lacking in focus.
- The experiments did not evaluate recent, well-aligned MLLMs, and the evaluation metrics were deemed insufficient for robust validation.
- The paper lacks critical analysis, such as justification for hyperparameter choices and thorough ablation studies.

The AC agrees that this work has potential but requires significant revision to address the major concerns raised by the reviewers for acceptance at a top-tier conference.

**Reviewer Concerns:**

Despite rebuttal efforts to address experimental issues, the fundamental limitation of this work is its incremental novelty.

**Reviewer Scores:**

While some scores might have seen minor adjustments, further discussion was unlikely to overturn the prevailing negative consensus among reviewers.

---

### Decision · Program_Chairs · 2026-01-26

Reject